# Impact of Age-Related Genetic Differences on the Therapeutic Outcome of Papillary Thyroid Cancer

**DOI:** 10.3390/cancers12020448

**Published:** 2020-02-14

**Authors:** Seok-Mo Kim, Soo Young Kim, Cheong Soo Park, Hang-Seok Chang, Ki Cheong Park

**Affiliations:** 1Thyroid Cancer Center, Gangnam Severance Hospital, Department of Surgery, Yonsei University College of Medicine, Seoul 120-752, Korea; SEOKMOKIM@yuhs.ac (S.-M.K.); KIMSUY@yuhs.ac (S.Y.K.); CSPARK1@yuhs.ac (C.S.P.); SURGHSC@yuhs.ac (H.-S.C.); 2Gangnam Severance Hospital, Department of Surgery, Yonsei University College of Medicine, 211, Eonjuro, Gangnam-gu, Seoul 135-720, Korea; 3Department of Surgery, Yonsei University College of Medicine, 50-1, Yonsei-ro, Seodaemun-gu, Seoul 120-752, Korea

**Keywords:** papillary thyroid carcinoma, EMT, FGFR, Lenvatinib, Sorafenib

## Abstract

The incidence of papillary thyroid carcinoma (PTC) has been increasing worldwide. PTC is the most common type of differentiated thyroid cancer and usually shows good prognosis. However, some PTC is driven to advanced stage by epithelial-mesenchymal transition (EMT)-mediated drug resistance, which is particularly noticeable in pediatric patients. There are limited options for systemic treatment, necessitating development of new clinical approaches. Here, we aimed to clarify genetic differences due to age of patients with PTC, and thereby aid in developing novel therapeutics. Patients with biochemically and histologically confirmed PTC were included in this study. PTC cells were acquired from young and older patients showing drug resistance, and were compared via microarray analysis. Cellular proliferation and other properties were determined after treatments with lenvatinib and sorafenib. In vivo, tumor volume and other properties were examined using a mouse xenograft model. Lenvatinib-treated group showed obvious suppression of markers of anti-apoptosis, EMT, and the FGFR signaling pathway, compared with control and Sorafenib-treated group. In the xenograft models, lenvatinib treatment induced significant tumor shrinkage and blocked the proto-oncogene Bcl-2 (B cell lymphoma/leukemia gene-2) and FGFR signaling pathway, along with reduced levels of EMT markers, compared with control and Sorafenib-treated group. Our findings clarify the age-dependent characteristics of pediatric PTC, giving insights into the relationship between young age and poor prognosis. Furthermore, it provides a basis for developing novel therapeutics tailored to the age at diagnosis.

## 1. Introduction

Thyroid carcinoma is the most common malignancy in adolescent and young adult women, and incidence is increasing across all age groups [1,2,3]. Thyroid carcinoma in children presents with several distinct clinical features when compared with that in adults. Despite a more advanced disease at presentation and a higher risk of recurrence, the prognosis of childhood differentiated thyroid carcinoma (DTC) is generally fairly good [4,5]. Biological and molecular mechanisms explaining the differing clinical behavior of pediatric and adult DTC have thus far been elusive. Several studies have suggested that the spectrum of mutations may differ between tumors of pediatric patients and those of adults [6,7,8]. Although there have been several previous studies describing molecular differences to explain the aggressive behavior of DTC in pediatric patients, no study has yet provided a clear explanation of the mechanism of action. As these aggressive pediatric cancers frequently acquire drug resistance, there is a need for effective clinical guidelines [9,10]. Recent studies have shown that the development of epithelial–mesenchymal transition (EMT) in advanced cancer cells not only results in metastasis, but also acts as a crucial contributing factor in fibroblast growth factor receptor (FGFR) signaling-mediated drug resistance [11,12]. Although the connection between EMT and drug resistance was confirmed a while ago, the mechanism remains unclear for advanced cancer. EMT is a physiological process wherein epithelial cells show collapse of cell-cell junctions and permanently transition to a state with the property of migratory cells [13]. Although EMT accounts for an essential aspect of resistance to ErbB-targeting compounds, a lack of understanding regarding the molecular mechanisms fundamental to this process has inhibited the progress of clinical approaches aimed at this drug-resistant state [14,15]. Some studies have shown that EMT and the FGFR signaling pathway play crucial roles in EMT-mediated poor prognosis and stemness of pediatric cancer [16,17,18,19]. These previous studies propose that stemness and aggressiveness of the pediatric cancer is related to EMT, which results in drug resistance, and consequently poor prognosis. Diverse mechanisms and molecules have been associated with poor clinical outcomes for advanced thyroid cancer [20,21]. In the current study, we focused on the relationship between drug resistance and EMT in pediatric thyroid cancer to explain the poor clinical outcomes [22]. Specifically, we investigated the potential underlying drug resistance mechanism, to aid the development of novel clinical approaches to address the issues related to drug resistance.

## 2. Results

### 2.1. Patients

The patients included 11 males and 41 females (Table 1), with a mean age at initial operation of 12.5 ± 2.9 years (range 5–15 years). A familial history of thyroid cancer was seen in 4 patients. The mean follow-up period was 9.8 ± 5.5 years. Cervical lymph node metastases were observed in nearly 80% of our cases. At initial presentation, two patients had distant lung metastasis. Recurrence was found in 9 (17.3%) of the 52 patients. TNM stage was defined using the AJCC Cancer Staging Manual, 8th Edition [23]. 

### 2.2. EMT Marker and FGFR Signaling Pathway Are Higher in Pediatric than in Adult Patients

To clarify the distinction between PTC of the young and older patients, we acquired four tissues (patients were treated at the Severance Hospital, Yonsei University College of Medicine, Seoul, Korea.) derived from 4 different patients with PTC (two young patients aged 4 and 13 and two old patients aged 47 and 52) (Table 2, Information for PTC from Severance Hospital; we decided to refer to the PTC in young age—under 15—as ‘GSPY’, and that in old age—over 40—as ‘GSPO’). Given that gene expression is largely dependent on age, we considered that GSPY achieves an aggressive phenotype by transcriptional reprogramming compared with that in GSPO.

To test this, we performed a gene expression microarray analysis to compare the GSPY and GSPO patient-derived PTC cells (Figure 1A). Many genes were significantly differentially expressed between GSPY and GSPO cells, suggesting that multiple biological processes were reprogrammed depending on age mediated cancer cell differentiation. Genes related to FGFR and EMT were particularly induced in GSPY (Figure 1A). Consistent with this, the protein expression related to FGFR and EMT was more dominant in GSPY than in GSPO (Figure 1B).

These data prove that genes and proteins related to the FGFR signaling pathway and EMT are expressed more highly in GSPY than in GSPO. Consequently, we focused on the mechanism by which aggressiveness was acquired via the FGFR signaling pathway and EMT.

### 2.3. Lenvatinib More Effectively Inhibited the Proliferation of GSPY than Sorafenib

To investigate the anti-cancer activities and IC_50_ of sorafenib and lenvatinib, we assayed cell proliferation in GSPY1 vs. GSPO1 and GSPY2 vs. GSPO2 in the presence and absence of these compounds using an MTT assay (Figure 2A–D, Table 3). The results indicated that both sorafenib and lenvatinib reduced the proliferation of GSPY and GSPO cells, compared with that of vehicle control-treated cells. However, sorafenib weakly suppressed the cell proliferation of GSPY compared with that of GSPO, with the difference being significant (Figure 2). Collectively, these results imply that lenvatinib is a more potent agent against GSPY.

### 2.4. Lenvatinib Induces More Effective Reduction of EMT-Mediated FGFR Signaling Pathway in Patient-Derived Thyroid Cancer Cells

Protein expression related to FGFR signaling pathway and EMT was more dominant in GSPY than in GSPO, this difference might account for the differences towards the anticancer drugs. Immunoblot analyses of protein levels in GSPY and GSPO cell lines indicated that lenvatinib showed marked decreases in the levels of PKC, MEK, p-ERK, Snail, and Zeb-1, which are well-known markers of FGFR signaling pathway and EMT, compared with that in response to sorafenib (Figure 3A). Notably, the anti-apoptotic marker Bcl-2 was highly suppressed in the lenvatinib treatment group compared with that in the sorafenib group. The TUNEL assay confirmed that lenvatinib induced apoptosis in GSPY and GSPO cell lines more potently than sorafenib (Figure 3B,C). Unlike sorafenib, lenvatinib is a multikinase inhibitor that inhibits FGFR signaling pathway, which effectively means the inhibition of EMT-mediated FGFR signaling pathway in GSPY. Together, these data indicate that lenvatinib effectively suppresses GSPO as well as GSPY.

### 2.5. Significant Tumor Shrinkage was Induced by Lenvatinib in a Xenograft Model of The Patient-Derived PTC Cells

To investigate the anti-cancer effects of sorafenib and lenvatinib in vivo, we developed a mouse xenograft tumor model with patient-derived PTC cells (GSPY1 vs. GSPO1, GSPY2 vs. GSPO2). Sorafenib used alone did not markedly suppress GSPY1 and 2 compared with GSPO1 and 2 cell xenograft tumors; however, lenvatinib resulted in tumor suppression in both GSPY- and GSPO-grafted mice (Figure 4A–D). Moreover, there was no evidence of systemic toxicity or treatment-related death in any group. Mouse body weight was not significantly influenced by treatment with sorafenib or lenvatinib (Figure 4E–H). The sorafenib and lenvatinib treatment groups had smaller tumor volumes than those in untreated mice. Interestingly, lenvatinib showed more significant tumor shrinkage than sorafenib on GSPY, unlike on GSPO (Figure 4I–L).

The lenvatinib treatment group also showed reduced levels of PKC, MEK, p-ERK, Snail, and Zeb-1 in GSPY and GSPO groups, compared with that after sorafenib treatment (Figure 5A). Notably, the lenvatinib treatment group also showed the stronger decrease in Bcl-2 expression compared with that after sorafenib treatment (Figure 5A,B). Together, these results imply that lenvatinib may be considered to have potent anti-cancer effects in GSPO as well as GSPY xenograft model.

## 3. Discussion

The clinical features of DTC are markedly different based on age, and this fact is probably associated with the differences in molecular profiles [24,25]. The high number of patients in whom no known genetic event was identified suggests that other genetic/epigenetic factors may be associated with the pathogenesis and biological behavior of pediatric DTC [26,27]. Cervical lymph node metastasis at diagnosis was observed in nearly 80% of our cases, while they were detected in 35% of adults in other studies [28,29,30]. The mechanism of action for aggressiveness in pediatric thyroid cancer has not been elucidated so far. Therefore, the current study attempted to reveal the reason for its aggressiveness in young age, unlike thyroid cancer in adults with respect to the EMT-mediated FGFR signaling pathway.

A gene expression microarray analysis was performed to compare young (GSPY) and old (GSPO) patient-derived papillary thyroid cancer cells. Various genes were significantly differentially expressed between GSPY and GSPO cases, indicating that EMT processes were reprogrammed depending on cancer cell aggressiveness and stemness. Genes related to EMT were significantly induced in GSPY than in GSPO. There has been much controversy over the appropriate treatment for adult patients and pediatric cases. Recognition and discovery of age-associated diagnostic markers, especially in pediatric patients with cancer would be significant in deciding a reasonable therapeutic approach. We were particularly interested in examining cancer cells isolated from pediatric patients who showed drug resistance [31]. A well-known study suggested that therapeutic failure in malignancies of pediatric cancer is due to drug resistance [32,33]. In cancer, EMT is associated with malignant progression, invasion, metastasis, and drug resistance [34,35,36,37]. CSCs (cancer stem cells) were found in poorly differentiated cancer or anaplastic thyroid cancer and have stem cell-like peculiarities [38,39,40,41] such as the ability for self-renewal, and were connected with metastasis, recurrence, and therapeutic resistance [42]. Recently, CSCs were associated with many cancers, including pediatric cancer [43]. Furthermore, a connection between EMT, drug resistance, and CSCs was recently confirmed [44]. Particularly, CSCs and cells showing EMT were considered to be crucial for drug resistance and metastasis, as observed in many human malignant or recurrent cancers [44]. Several studies have also indicated the relationship between EMT and drug resistance in CSCs [15,44,45]. Interestingly, pediatric cancers have properties similar to CSCs with respect to EMT-mediated stemness.

In the current study, in vivo experiments using a mouse xenograft tumor model indicated that sorafenib alone could not markedly suppress GSPY compared GSPO cases; however, lenvatinib treatment resulted in tumor suppression in both GSPY and GSPO cases. Our findings indicate that the drug resistance in pediatric PTC, i.e., sorafenib-resistant patient-derived PTC cell, could be inhibited by lenvatinib, through the inhibition of the EMT-mediated FGFR signaling pathway and Bcl-2. It is a well-known fact that the anti-apoptotic factor and proto-oncogene Bcl-2 is a key player in the control of apoptosis [46,47,48]. These results resolutely showed that Bcl-2 is also involved in EMT, an essential mechanism in the drug resistance of pediatric PTC [49,50]. These findings can be useful to design future rational clinical studies on pediatric patient with drug resistant cancer cell in order to develop effective therapies. In microarray experiments, however, it is thought that there will be more signaling pathways involved in these outcomes. Further studies are needed to assess current outcomes. Furthermore, more research will be needed next study due to the limitation of only four patient outcomes in this study.

It is noteworthy that these findings suggest that the therapeutic trials based on genetics, such as those involving the inhibition of the EMT-mediated FGFR signaling pathway, are a potentially effective new clinical approach for the care of pediatric patients with CSCs having drug-resistant properties.

## 4. Materials and Methods

### 4.1. Patients

The patients recruited for this study underwent thyroid operation, including thyroidectomy and cervical lymph node dissection, and were ≤15 years of age at the time of surgery. They were treated from March 2003 to December 2015 at the Gangnam Severance Hospital (Seoul, South Korea). The details of the patients’ presentations, family history, surgical and pathological findings, and outcome were obtained from the database of Thyroid Cancer Center.

### 4.2. Patient Tissue Specimens

Fresh tumors were acquired from patients with biochemically and histologically established PTC who were treated at the Severance Hospital, Yonsei University College of Medicine, Seoul, Korea. Fresh tumors were obtained during surgical resection of thyroid cancer primary and metastatic sites. Some patients with thyroid cancer were chosen depending on cancer subtype. The research protocol was approved by the Institutional Review Board of the Thyroid Cancer Center, Gangnam Severance Hospital, Yonsei University College of Medicine (IRB Protocol: 3-2019-0281, Cell samples were acquired from patients at the Severance Hospital, Yonsei University College of Medicine, Seoul, Korea).

### 4.3. Tumor Cell Isolation and Primary Culture

After resection, tumors were kept in normal saline with antifungal and antibiotics and moved to the laboratory. Normal tissue and fat were removed, and the tissues were rinsed with 1× Hank’s Balanced Salt Solution. Tumors were minced in a tube with dissociation medium containing Dulbecco’s Modified Eagle’s Medium/Nutrient Mixture F-12 (DMEM/F12) with 20% fetal bovine serum (FBS) supplemented with 1 mg/mL collagenase type IV (Sigma, St. Louis, MO, USA; C5138). The minced and suspended tumor cells were filtered through sterile nylon cell strainers with 70-micron pores (BD Falcon, Franklin Lakes, NJ, USA), rinsed with 50 mL of 1× Hank’s Balanced Salt Solution, and centrifuged at 220× *g* for 5 min. Cells were re-suspended in RPMI-1640 (Hyclone, South Logan, UT, USA) medium with 10% FBS (Hyclone) and 2% penicillin/streptomycin solution (Gibco, Grand Island, NY, USA). Cell viability was determined using the trypan blue dye exclusion method.

### 4.4. Cell Culture

The patient-derived PTC cells were isolated and grown in RPMI-1640 medium with 10% FBS (cells were authenticated by short tandem repeat profiling, karyotyping, and isoenzyme analysis).

### 4.5. Cell Proliferation Assay

Cell proliferation was measured using the MTT assay. Cells were seeded in 96-well plates at 6 × 10^3^ cells per well and incubated overnight to achieve 80% confluency. Cells were incubated for the indicated times prior to the determination of cell proliferation using the MTT reagent (Roche, Basel, Switzerland; 11465007001) according to the manufacturer’s protocol. Absorbance was measured at 550 nm. Viable cells were counted by trypan blue exclusion. Data were calculated as a percentage of the signal observed in vehicle-treated cells and are shown as the means ± SEM of triplicate experiments.

### 4.6. Microarray Experiment and Data Analysis

RNA purity and integrity were evaluated using an ND-1000 Spectrophotometer (NanoDrop, Wilmington, DE, USA) and an Agilent 2100 Bioanalyzer (Agilent Technologies, Palo Alto, CA, USA). RNA labeling and hybridization were performed using the Agilent One-Color Microarray-Based Gene Expression Analysis protocol (Agilent Technology, V 6.5, 2010). Briefly, 100 ng of total RNA from each sample was linearly amplified and labeled with Cy3-dCTP. The labeled cRNAs were purified using an RNeasy Mini Kit (Qiagen, Venlo, The Netherlands). The concentration and specific activity of the labeled cRNAs (pmol Cy3/μg cRNA) were measured using the NanoDrop ND-1000. Then, 600 ng of each labeled cRNA was fragmented by adding 5 μL of 10× blocking agent and 1 μL of 25× fragmentation buffer, and then heated at 60 °C for 30 min. Finally, 25 μL of 2× GE hybridization buffer was added to dilute the labeled cRNA. Hybridization solution (40 μL) was dispensed into the gasket slide and assembled to the Agilent SurePrint G3 Human GE 8X60K, V3 Microarrays (Agilent^®^). Raw data were extracted using Agilent Feature Extraction Software (v11.0.1.1). The raw data for each gene were then summarized automatically in an Agilent feature extraction protocol to generate the raw data text file, providing expression data for each gene probed on the array. Gene-enrichment and functional annotation analysis for the significant probe list was performed using gene ontology (www.geneontology.org/) and Kyoto Encyclopedia for Genes and Genomes (http://kegg.jp) analyses. All data analysis and visualization of the differentially expressed genes were conducted using R 3.1.2 (www.r-project.org).

### 4.7. Evaluation of Apoptotic Cell Death

Cells were fixed with 4% paraformaldehyde solution for 48 h and then analyzed using a terminal deoxynucleotidyl transferase dUTP nick end labeling (TUNEL) kit (Promega, Madison, WI, USA). The apoptotic cells (fluorescent green) and total cells were counted by fluorescence microscopy. Images were acquired under a confocal microscope (LSM Meta 700, Carl Zeiss, Oberkochen, Germany) and analyzed using the Zeiss LSM Image Browser software, version 4.2.0121.

### 4.8. Immunoblot Analysis

Cells were washed twice with cold phosphate-buffered saline and lysed on ice with protein extraction buffer (Pro-Prep, iNtRON Biotechnology, Seoul, Korea) following the manufacturer’s protocol. Protein concentrations were determined by a BCA assay (Pierce Biotechnology, Rockford, IL, USA). Equal amounts of protein (20 μg) were separated on 8–10% sodium dodecyl sulfate-polyacrylamide gels; the resolved proteins were electro-transferred onto polyvinylidene fluoride membranes (Millipore, Bedford, MA, USA). The membranes were subsequently blocked with 5% nonfat milk in TBST for 1 h at room temperature and incubated with appropriate concentrations of primary antibodies against PKC, MEK, Snail, Zeb1,2 (all from Abcam, Cambridge, UK) and p-ERK1/2, ERK1/2, Bcl-2, β-actin (all from Santa Cruz Biotechnology, Dallas, TX, USA) overnight at 4 °C. The membranes were then rinsed 3–5 times with TBST and probed with the corresponding secondary antibodies conjugated to horse radish peroxidase (Santa Cruz) at room temperature for 1 h. After rinsing, the blots were developed with ECL reagents (Pierce) and exposed using Kodak X-OMAT AR Film (Eastman Kodak, Rochester, NY, USA) for 3–5 min. Uncropped images of the immunoblot analysis indicated on Appendix A).

### 4.9. Immunohistochemistry

All tissues were fixed in 10% neutral-buffered formalin and embedded in paraffin wax following standard protocols. Tissue sections (5 μm) were dewaxed, and antigen retrieval was performed in citrate buffer (pH 6), using an electric pressure cooker set at 120 °C for 5 min. Sections were incubated for 5 min in 3% hydrogen peroxide to quench endogenous tissue peroxidase. Primary monoclonal antibodies against Bcl-2 (Abcam) was diluted with PBS at a ratio of 1:100 and incubated overnight at 4 °C. All tissue sections were counterstained with hematoxylin, dehydrated, and mounted.

### 4.10. Image Analysis

MetaMorph 4.6 software (Universal Imaging Co., Downington, PA, USA) was used for computerized quantification of immunostained target proteins.

### 4.11. Human Papillary Thyroid Cancer Cell Xenografts

The patient-derived PTC cells (4.0 × 10^6^ cells/mouse) were cultured in vitro and then injected subcutaneously into the upper left flank region of 5- to 6-week-old female BALB/c nude mice. After 13 days, tumor-bearing mice were assigned to groups randomly (*n* = 10/group) and administered 10 mg/kg lenvatinib (p.o.) and 40 mg/kg sorafenib (p.o.), once every 2 days. Tumor size was measured every other day using calipers. Tumor volume was estimated using the following formula: L × S2/2 (L, longest diameter; S, shortest diameter). Animals were maintained under specific pathogen-free conditions. All experiments were approved by the Animal Experiment Committee of Yonsei University.

### 4.12. Statistical Analysis

Statistical analyses were performed using GraphPad Prism 6.0 (GraphPad Software Inc., La Jolla, CA, USA). Immunohistochemistry results were evaluated by ANOVA followed by Bonferroni post hoc tests. Values are expressed as the means ± standard deviation (SD). *p* < 0.05 indicated statistical significance.

## 5. Conclusions

In conclusion, our study proposed that Bcl-2 is also involved in EMT, an essential mechanism in the drug resistance of pediatric PTC. Consequentially, these findings can be useful to design future rational clinical studies on pediatric patient with drug resistant cancer cell in order to develop effective therapies.

## Figures and Tables

**Figure 1 cancers-12-00448-f001:**
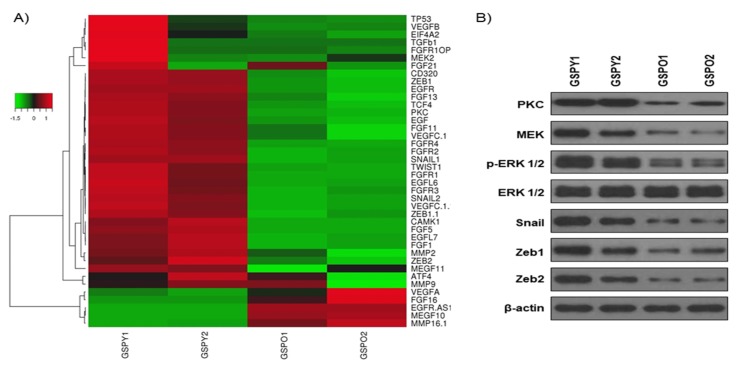
Gene expression profiles of patient-derived papillary thyroid cancer (PTC) cells. Gene expression analysis using a microarray approach between PTC of young (GSPY) and old (GSPO) individuals. (**A**) Gene expression profiles based on microarray. Hierarchical clustering analysis for comparison of GSPY and GSPO samples. (**B**) Immunoblot analysis for markers of FGFR signaling pathway and EMT in GSPY and GSPO samples.

**Figure 2 cancers-12-00448-f002:**
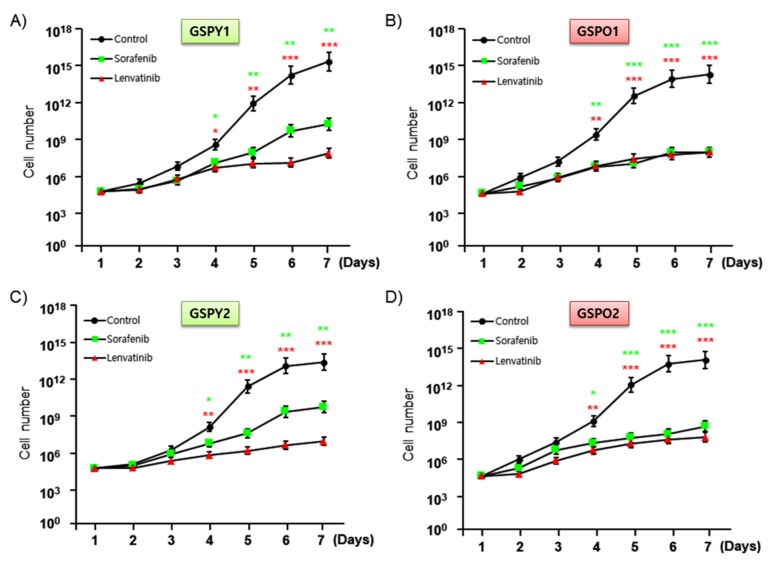
Anti-cancer effects of lenvatinib or sorafenib on patient-derived papillary thyroid cancer cells from young (GSPY) and older (GSPO) patients. Cell proliferation assay after treatments with lenvatinib or sorafenib alone: (**A**,**C**) GSPY; (**B**,**D**) GSPO. The data points were calculated as mean % of the value observed in the solvent-treated control. All experiments were repeated at least 3 times. Data are represented as means ± SD. * *p* < 0.05, ** *p* < 0.01, and *** *p* < 0.005 compared with control.

**Figure 3 cancers-12-00448-f003:**
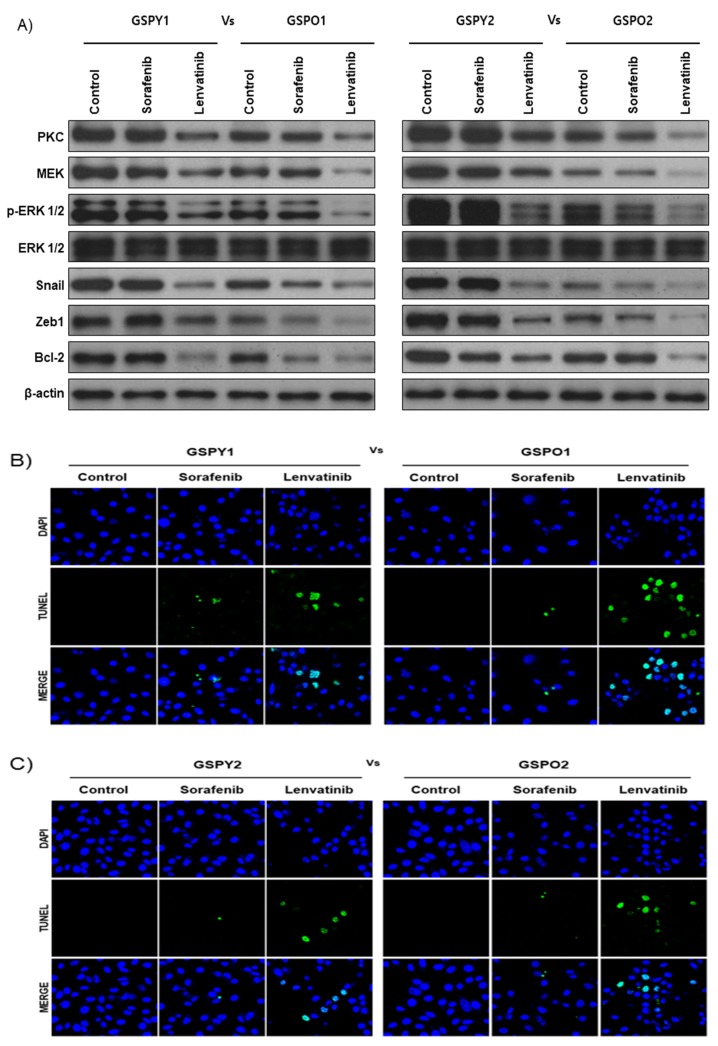
The expression of markers associated with the FGFR signaling pathway, EMT, and anti-apoptosis were significantly suppressed by lenvatinib compared with that by sorafenib. (**A**) Immunoblot analysis for markers of FGFR signaling pathway (PKC, MEK and p-ERK1/2), EMT (Snail and Zeb1), and anti-apoptotic factor (Bcl-2) in young (GSPY) and older (GSPO) patient-derived papillary thyroid cancer cells. (**B**,**C**) TUNEL assay of GSPY and GSPO; TUNEL-positive (apoptotic) cells are indicated (×400): (**B**) GSPY1 vs. GSPO1; (**C**) GSPY2 vs. GSPO2.

**Figure 4 cancers-12-00448-f004:**
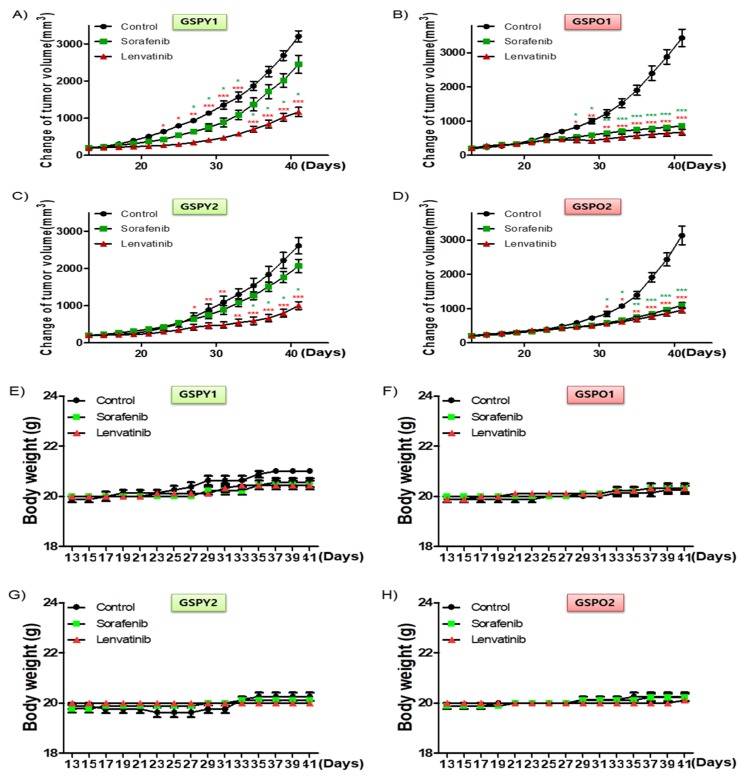
Tumor-selective anti-cancer effects of lenvatinib or sorafenib in patient-derived papillary thyroid cancer cell xenografts in vivo. Athymic nude mice with established tumors were treated with the indicated inhibitors. Data represent the mean tumor volumes. Inhibition of tumor progression by therapy with lenvatinib or sorafenib in mice grafted with patient-derived papillary thyroid cancer cell: (**A**,**E**,**I**) GSPY1; (**B**,**F**,**J**) GSPO1; (**C**,**G**,**K**) GSPY2; (**D**,**H**,**L**) GSPO2 (*n* = 10 mice/group). (**A**–**D**) Change in tumor volume. (**E**–**H**) The compounds had no significant effect on mouse body weight. (**I**–**L**) Weight of the dissected tumors. * *p* < 0.05, ** *p* < 0.01, and *** *p* < 0.005 compared with the control.

**Figure 5 cancers-12-00448-f005:**
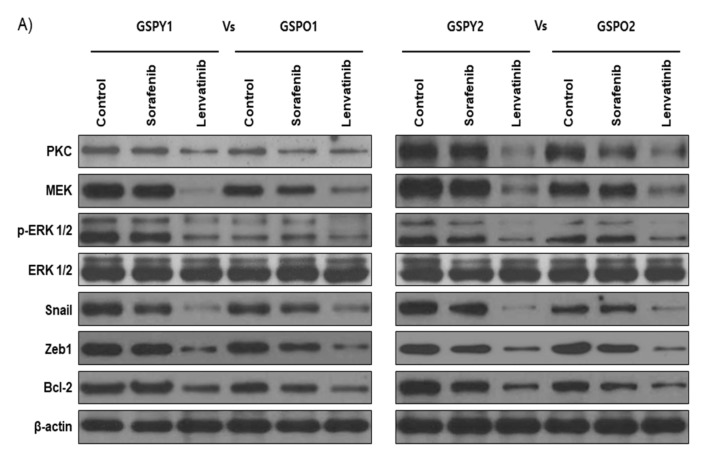
(**A**) Immunoblot analysis of FGFR signaling pathway, EMT, and anti-apoptotic-related proteins in tumor tissues derived from xenograft models using young (GSPY) and older (GSPO) patient-derived papillary thyroid cancer cells. (**B**) Immunohistochemical analysis of Bcl-2 protein levels in paraffin-embedded xenograft tumor tissues. Scale bar, 80 μm. Lenvatinib and sorafenib were administered orally (p.o.), once every 2 days and total lysates or paraffin-embedded were prepared from GSPY and GSPO xenograft model. The assay estimated the marker levels of FGFR signaling pathway (PKC, MEK and p-ERK1/2), EMT (Snail and Zeb1), and anti-apoptotic factor (Bcl-2) in GSPY and GSPO samples. * *p* < 0.05, ** *p* < 0.01, and *** *p* < 0.005 compared with the control.

**Table 1 cancers-12-00448-t001:** Clinical and pathological characteristics of thyroid carcinoma in 52 patients aged 15 years old and less at diagnosis. SD, standard deviation.

Variables	Patients
	N = 52
Age (years), mean ± SD (range)	12.5 ± 2.9 (5–15)
Male:female ratio, n (%)	11 (21.2):41 (78.8)
Initial presentation, n (%)	
Incidentaloma	18 (34.6%)
Palpable mass	33 (63.5%)
Thyrotoxicosis	1 (1.9%)
Famliy history of thyroid cancer, n (%)	4 (7.7%)
Histology, n (%)	
Papillary	45 (86.5%)
Follicular	6 (11.5%)
Medullary	1 (1.9%)
Tumor size (cm), mean ± SD	2.4 ± 1.4
Extrathyroidal extension	32 (61.5%)
Vascular invasion	41 (78.8%)
TNM stage	
T stage	
T1:T2:T3:T4	13 (25.0):4 (7.7):31 (59.6):4(7.7)
N stage	
N0:N1a:N1b	12 (23.1):9 (17.3):31 (59.6)
M stage	
M0:M1	50 (96.2):2 (3.8)
Time to recurrence (years), mean ± SD (range)	8.3 ± 5.6 (1.1–11.7)
Follow up period (years), mean ± SD (range)	9.8 ± 5.5 (4.1–13.7)

**Table 2 cancers-12-00448-t002:** Characteristics of patient-derived papillary thyroid cancer cell line, including viability after drug treatment of all thyroid cancer cell lines examined.

Variables	GSPY1	GSPY2	GSPO1	GSPO2
**Age at Diagnosis**	4	13	52	47
**Gender**	Female	Female	Female	Female
**Primary Disease Site**	Thyroid	Thyroid	Thyroid	Thyroid
**Stage**	T4aN1bM0	T4aN1bM0	T4aN1bM1	T3N1bM1
**Primary Pathology (subtype)**	Papillary thyroid cancer (conventional)	Papillary thyroid cancer (conventional)	Papillary thyroid cancer (conventional)	Papillary thyroid cancer (conventional)
**Classification of specimen used for culture**	Fresh tumor	Fresh tumor	Fresh tumor	Fresh tumor
**Obtained from**	Gangnam Severance Hospital, Seoul, Korea	Gangnam Severance Hospital, Seoul, Korea	Gangnam Severance Hospital, Seoul, Korea	Gangnam Severance Hospital, Seoul, Korea

**Table 3 cancers-12-00448-t003:** IC_50_ based on cell proliferation assay. Each data point represents the mean of 3 independent MTT assays; IC_50_ was determined in triplicate. SD, standard deviation.

Cell Line	Hisopathology	Animal	Cell Proliferation IC_50_ (μM)	
			Sorafenib	Lenvatinib
**GSPY1**	Thyroid cancer; Papillary	Human	17.24 (±0.5)	15.75 (±0.2)
**GSPY2**	Thyroid cancer; Papillary	Human	20.25 (±0.1)	16.54 (±0.3)
**GSPO1**	Thyroid cancer; Papillary	Human	9.84 (±0.5)	10.23 (±0.4)
**GSPO2**	Thyroid cancer; Papillary	Human	10.12 (±0.2)	10.98 (±0.1)

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
