# Peer review of "Impact of Age-Related Genetic Differences on the Therapeutic Outcome of Papillary Thyroid Cancer"

_cancers, 2020, doi:10.3390/cancers12020448_

Round 1
Reviewer 1 Report
The manuscript is well written and data presented in a logical flow. There are a few minor comments:
Line 101: 2 young and 2 old age patients - please specify how old were these four patients Figure 5B: This is not clear what actually this IHC pictures present. Additionally to the treatment and patient sample the pictures should indicate which marker/antibody was used. Line 245: "is " is missing before "key player"Author Response
Reviewer 1
Line 101: 2 young and 2 old age patients - please specify how old were these four patients Reply: Thank you for your comment. I agree with you completely. I have made the suggested correction. The text now reads as “two young patients aged 4 and 13 and two old patients aged 47 and 52”, line 126.Figure 5B: This is not clear what actually this IHC pictures present. Additionally to the treatment and patient sample the pictures should indicate which marker/antibody was used. Reply: Thank you for your comment. I have indicated marker/antibody used in Figure 5B. Line 245: "is " is missing before "key player" Reply: Thank you for your comment. I have made this correction, line 274.

Reviewer 2 Report
Review of the article entitled:
Genetic Basis Mediated Therapy Trials for Between Age and Papillary Thyroid Cancer
The authors present a study in which they evaluate a therapy based on genetic research, compared between pediatric and adult patients with PTC
Comments:
some minor language mistakes should be corrected, maybe it would be beneficial for the work to rephrase the title, because I could understand it only after reading the manuscript.Other lines that need English editing:
line 245 – “which key player in control of the apoptosis”
Line 251 – “It is a noteworthy, these findings suggest that the genetic basis mediated-therapy trials, inhibition of EMT-mediated FGFR signaling pathway, are a potentially effective new clinical approach for the care of pediatric patients with CSCs with drug-resistant properties”
Table 1 - is the information available on histologic subtypes of PTC, extrathyroidal extension, vascular invasion?Table 2 – different variants of cancer in PTC patients may have different prognosis. Any additional information could be useful to determine similarities among the patients used for comparisons, to exclude that any other factors different than age could influence the results (histologic subtypes of PTC, extrathyroidal extension especially in patient GSPO2), especially if you compare only four patients.
Table 1, table 2 - Which edition of TNM classification do you use for cancer staging?
Table 2 – please provide a legend
I understand that some data may not be available, but in this case it should be mentioned as study limitations.
lines 214 – “Cervical lymph node metastasis at diagnosis was observed in nearly 80% of our cases, while they were detected in 35% of adults in other studies [27-29]” – did you mean 35% of ADULTS or perhaps 35% of CHILDREN? The authors compare only four patients in their study. This obviously results from the rarity of therapy with kinase inhibitors, but in my opinion the fact that the study describes only four patients should also be mentioned as a study limitation.Author Response
Reviewer 2
Some minor language mistakes should be corrected, maybe it would be beneficial for the work to rephrase the title, because I could understand it only after reading the manuscript. Reply: Thank you for this comment. I have changed the title to reflect the contents of the manuscript better. Other lines that need English editing:- line 245 – “which key player in control of the apoptosis”
- Line 251 – “It is a noteworthy, these findings suggest that the genetic basis mediated-therapy trials, inhibition of EMT-mediated FGFR signaling pathway, are a potentially effective new clinical approach for the care of pediatric patients with CSCs with drug-resistant properties”
Reply: Thank you for your comment. I have corrected these sections, line 280. Table 1 - is the information available on histologic subtypes of PTC, extrathyroidal extension, vascular invasion? Reply: Thank you for your comments. I have included this information in Table 1.Table 2 – different variants of cancer in PTC patients may have different prognosis. Any additional information could be useful to determine similarities among the patients used for comparisons, to exclude that any other factors different than age could influence the results (histologic subtypes of PTC, extrathyroidal extension especially in patient GSPO2), especially if you compare only four patients. Reply: Thank you for your comment. I agree with you. Although the T stage is different for GSPO2, extrathyroidal extension is common to all four patients. All histologic subtypes of PTC were conventional. Pediatric papillary cancer was not the diffuse sclerosing type that is commonly observed in pediatric thyroid cancer. If the M1 stage was common to all four patients, better results would have been obtained; however, it was very difficult to obtain samples of pediatric patients with an M1 stage of papillary thyroid cancer. Nevertheless, it is meaningful that the cancer was more aggressive in GSPY1,2, which have a lower stage than GSPO1,2.
Table 1, table 2 - Which edition of TNM classification do you use for cancer staging? Reply: Thank you for your comments. I have included “TNM stage was defined using the AJCC cancer staging manual 8th in section 2.1 of the Results.
Table 2 – please provide a legend Reply: Thank you for your comment. I have added the legend for Table 2.
I understand that some data may not be available, but in this case it should be mentioned as study limitations. - lines 214 – “Cervical lymph node metastasis at diagnosis was observed in nearly 80% of our cases, while they were detected in 35% of adults in other studies [27-29]” – did you mean 35% of ADULTS or perhaps 35% of CHILDREN? Reply: Thank you for your comment. It is meaning the CHILDREN. One of the indicators of cancer aggressiveness, cervical metastasis, is observed more in children than in adults. The authors compare only four patients in their study. This obviously results from the rarity of therapy with kinase inhibitors, but in my opinion the fact that the study describes only four patients should also be mentioned as a study limitation. Reply: Thank you for your comments. I agree with you. This study may be limited. Thus, we're going to compare more patients in the next research. And I mentioned to ‘Discussion’ in the manuscript, line 280.

Round 2
Reviewer 2 Report
I have deleted the commentaries which have been satisfactorily addressed and left below two comments that still need clarification:
Reviewer 2
Some minor language mistakes should be corrected, maybe it would be beneficial for the work to rephrase the title, because I could understand it only after reading the manuscript. Reply: Thank you for this comment. I have changed the title to reflect the contents of the manuscript better.Thank You for taking into account my suggestions as far as the title is concerned, however I’m afraid the title still needs clarification. I understand that it is not my role to correct the language, but please consider using “impact” or “effect” instead of “affect”.
Table 2 – different variants of cancer in PTC patients may have different prognosis. Any additional information could be useful to determine similarities among the patients used for comparisons, to exclude that any other factors different than age could influence the results (histologic subtypes of PTC, extrathyroidal extension especially in patient GSPO2), especially if you compare only four patients. Reply: Thank you for your comment. I agree with you. Although the T stage is different for GSPO2, extrathyroidal extension is common to all four patients. All histologic subtypes of PTC were conventional. Pediatric papillary cancer was not the diffuse sclerosing type that is commonly observed in pediatric thyroid cancer. If the M1 stage was common to all four patients, better results would have been obtained; however, it was very difficult to obtain samples of pediatric patients with an M1 stage of papillary thyroid cancer. Nevertheless, it is meaningful that the cancer was more aggressive in GSPY1,2, which have a lower stage than GSPO1,2.
Please consider adding to the table information about the subtype of thyroid cancer
Author Response
Some minor language mistakes should be corrected, maybe it would be beneficial for the work to rephrase the title, because I could understand it only after reading the manuscript. Reply: Thank you for this comment. I have changed the title to reflect the contents of the manuscript better.Thank You for taking into account my suggestions as far as the title is concerned, however I’m afraid the title still needs clarification. I understand that it is not my role to correct the language, but please consider using “impact” or “effect” instead of “affect”. Reply: I don't know how to thank you enough. I have changed to “Affect” → “Impact” in the title. Table 2 – different variants of cancer in PTC patients may have different prognosis. Any additional information could be useful to determine similarities among the patients used for comparisons, to exclude that any other factors different than age could influence the results (histologic subtypes of PTC, extrathyroidal extension especially in patient GSPO2), especially if you compare only four patients. Reply: Thank you for your comment. I agree with you. Although the T stage is different for GSPO2, extrathyroidal extension is common to all four patients. All histologic subtypes of PTC were conventional. Pediatric papillary cancer was not the diffuse sclerosing type that is commonly observed in pediatric thyroid cancer. If the M1 stage was common to all four patients, better results would have been obtained; however, it was very difficult to obtain samples of pediatric patients with an M1 stage of papillary thyroid cancer. Nevertheless, it is meaningful that the cancer was more aggressive in GSPY1,2, which have a lower stage than GSPO1,2.Please consider adding to the table information about the subtype of thyroid cancer Reply: Thank you for your comments, I added information about the subtype of thyroid cancer.